# Endothelial Cells Tissue-Specific Origins Affects Their Responsiveness to TGF-β2 during Endothelial-to-Mesenchymal Transition

**DOI:** 10.3390/ijms20030458

**Published:** 2019-01-22

**Authors:** Fernanda Ursoli Ferreira, Lucas Eduardo Botelho Souza, Carolina Hassibe Thomé, Mariana Tomazini Pinto, Clarice Origassa, Suellen Salustiano, Vitor Marcel Faça, Niels Olsen Câmara, Simone Kashima, Dimas Tadeu Covas

**Affiliations:** 1Center for Cell Based Therapy, Regional Blood Center, Ribeirão Preto 14051-140, Brazil; lucas.edubs@gmail.com (L.E.B.S.); carolhthome@gmail.com (C.H.T.); marianatomazini@yahoo.com (M.T.P.); suellengsalustiano@gmail.com (S.S.); skashima@hemocentro.fmrp.usp.br (S.K.); dimas@fmrp.usp.br (D.T.C.); 2Institute of Biomedical Sciences, University São Paulo, São Paulo 05508-900, Brazil; clarice.jp@gmail.com (C.O.); niels.camara@gmail.com (N.O.C.); 3Department of Biochemistry and Immunology, FMRP-University of São Paulo, Ribeirão Preto, SP 14049-900, Brazil; vmfaca@gmail.com; 4Department of Internal Medicine, FMRP-University of São Paulo, Ribeirão Preto, SP 14049-900, Brazil

**Keywords:** endothelial cells, TGF-β2, endothelial-mesenchymal transition, epithelial-mesenchymal transition, signaling pathways

## Abstract

The endothelial-to-mesenchymal transition (EndMT) is a biological process where endothelial cells (ECs) acquire a fibroblastic phenotype after concomitant loss of the apical-basal polarity and intercellular junction proteins. This process is critical to embryonic development and is involved in diseases such as fibrosis and tumor progression. The signaling pathway of the transforming growth factor β (TGF-β) is an important molecular route responsible for EndMT activation. However, it is unclear whether the anatomic location of endothelial cells influences the activation of molecular pathways responsible for EndMT induction. Our study investigated the molecular mechanisms and signaling pathways involved in EndMT induced by TGF-β2 in macrovascular ECs obtained from different sources. For this purpose, we used four types of endothelial cells (coronary artery endothelial cells, CAECs; primary aortic endothelial cells PAECs; human umbilical vein endothelia cells, HUVECs; and human pulmonary artery endothelial cells, HPAECs) and stimulated with 10 ng/mL of TGF-β2. We observed that among the ECs analyzed in this study, PAECs showed the best response to the TGF-β2 treatment, displaying phenotypic changes such as loss of endothelial marker and acquisition of mesenchymal markers, which are consistent with the EndMT activation. Moreover, the PAECs phenotypic transition was probably triggered by the extracellular signal–regulated kinases 1/2 (ERK1/2) signaling pathway activation. Therefore, the anatomical origin of ECs influences their ability to undergo EndMT and the selective inhibition of the ERK pathway may suppress or reverse the progression of diseases caused or aggravated by the involvement EndMT activation.

## 1. Introduction

The vascular system is a complex network of vessels that connects the heart to several tissues and organs in order to maintain homeostasis in response to physiological and pathological changes. Endothelial cells (ECs), which are anatomically similar to the squamous epithelium, have the apical-basal polarity and are strongly linked together by tight junctions and adherents junctions [1]. These cells are located in the inner surface of blood and lymphatic vessels and play important roles in the development and remodelling of the vasculature, organs, maintenance of vascular tone, blood flow, coagulation, and exchange of nutrients [2,3].

The diversity of the vasculature, in terms of haemodynamics, structure, and embryonic origin, requires that their constituent ECs also have regional specialties in morphology and physiological functions [4]. ECs heterogeneity has been described at the level of cell morphology, function, gene expression, and antigen composition [5]. For example, using microarray analysis, Chi and colleagues demonstrated that macrovascular and microvascular ECs of different anatomical locations, grown under the same conditions, have distinct gene expression profiles [4].

The endothelium can be affected in several ways, the most notable way is possibly an endothelial extreme plasticity form known as endothelial-to-mesenchymal transition (EndMT). During this process, ECs assume a mesenchymal phenotype and migrate towards the underlying tissue. This phenotypic change is characterized by the loss of cell-cell junctions, acquisition of invasive and migratory properties, loss of endothelial markers, such as platelet endothelial adhesion molecule PECAM-1 (CD31), VE-cadherin, and von Willebrand factor (vWF). Concomitantly, ECs upregulates the expression of mesenchymal markers, such as fibroblast specific protein 1 (FSP-1), N-cadherin, fibronectin, and vimentin; and also marks mural cells like alpha-smooth muscle actin (α-SMA) and calponin [1,6,7,8].

ECs plasticity is crucial in physiological processes, such as development and regeneration, as well as in pathological processes, such as fibrosis and tumour progression. This process is mainly involved during embryonic development of the heart, where the mesenchymal cells that form the atrioventricular cushion, the primordia of the valves and septa of the adult heart, are derived from the endocardium by EndMT [9,10,11,12].

Several signalling pathways have been implicated in physiological and pathological activation of EndMT, such as Notch, canonical Wnt and TGF-β pathways [13,14,15]. However, the EndMT seems to be mainly activated by TGF-β2 isoform [16,17]. The EndMT response by TGF-β induction in ECs occurs through pathways both dependent and independent of Smad, such as via mitogen-activated protein kinases/extracellular signal-regulated kinases-MEK (MAPK/ERK), phosphoinositide 3-kinase (PI3K) and p38 MAPK [16].

Although it is known that TGF-β participates in important biological processes of ECs, as in cell ontogeny, is not clear yet how ECs from distinct sources respond to the induction of EndMT by TGF-β2 treatment. The aim of this study was to analyse the effect of TGF-β2 in endothelial cells from different sources, and to elucidate signalling pathways involved in EndMT. Such knowledge can be beneficial for the design of target therapies against pathological EndMT activation in specific vascular beds or specific pathologies like fibrosis, cancer, diabetes, and atherosclerosis.

In this study, we observed that TGF-β2 induces EndMT in vitro in different types of ECs. However, ECs responded differently. Among the macrovascular ECs, aortic artery endothelial cells were the most responsive to TGF-β2, presenting morphological and molecular changes consistent with the in vitro EndMT process. The changes observed in these cells were probably initiated by the activation of the Erk1/2 pathway. Thus, future studies would be necessary to the selective inhibition of the Erk pathway in order to suppress pathologies induced and aggravated by TGF-β2.

## 2. Results

The obtained results in this study demonstrated that ECs isolated from different anatomical sites behave differently during TGF-β2-mediated activation of EndMT. This heterogeneity may result in differential response against physiological and pathological EndMT-triggering signals.

One of the major concerns in using primary cells in experiments is the degree of contaminants with other cell types. Thus, all cells types were properly characterized before performing each procedure. The EC population was obtained after four culture passages, when they presented a uniform and typically polygonal morphological appearance typical of CEs (Figure 1A). Based on data obtained after flow cytometric immunophenotyping, ECs expressed little or none of the CD34 (Cluster of Differentiation 34), CD45, and CD14 hematopoietic cell antigens, nor the typical CD90 mesenchymal antigen, whereas they were positive for endothelial markers such as CD31, CD144, CD105, CD146, among others (Figure 1B). In addition, the functional test of the EC cultures in matrigel, showed the ability of ECs to form complex structures similar to vascular networks, evidencing characteristics typical of CEs (Figure 1C). After five days of treatment with TGF-β2, only PAEC (but not CAEC, HPAEC, and HUVEC) exhibited alterations in the cell morphology, from endothelial cobblestone-like to an elongated spindle-shaped form, which is a characteristic feature of the mesenchymal cells (Figure 2).

In consequence, we investigated the transcriptional profile of EndMT markers after TGF-β2 treatment. Real-time quantitative PCR (qPCR) analysis revealed distinct transcriptional profiles among ECs. Regarding HUVECs, TGF-β2 incubation had little effect on mesenchymal markers expression, since only actin, alpha cardiac muscle 1 (*ACTC1*) was upregulated (27-fold) after treatment (Figure 3A). In contrast, HPAECs showed increased transcription levels of mesenchymal markers such as *SM-22α*, *ACTC1*, and calponin 1 (*CNN1*) (2.3-fold, 22-fold, and 10-fold, respectively) including *SNAIL* (1.7-fold), which is a transcriptional factor involved in EndMT activation. CAECs demonstrated upregulation of collagen type 1 (*Col1A1*), *ACTC1* and *CNN1* (8-fold, 24-fold and 2-fold, respectively) transcription levels (Figure 3B,C). Of note, TGF-β2 treatment of PAECs induced the strongest upregulation of *Col1A1* (~290 fold increase) along with the expression of other mesenchymal markers: *SM-22α*, *ACTC1* and *CNN1* (5-fold, 5-fold, and 15-fold increase, respectively). In addition, only these cells exhibited an increase of *SLUG* mRNA (3-fold), another transcriptional factor that is involved in EndMT activation (Figure 3D). Only PAECs, after treatment with TGF-β2, showed increased SM-22α at the protein level (Figure 3E) which is in accordance with the most pronounced EndMT transcriptional profile.

Despite the upregulation of mesenchymal markers, the transcription levels of the endothelial marker *CD31* were not suppressed in any of the treated ECs (Figure 3A–D). However, immunofluorescence staining of TGF-β2-treated cells showed that CD31 was downregulated in PAEC, CAEC, and HUVEC, but not in HPAEC (represented by green fluorescence). Remodelling of actin filaments is necessary for EndMT. Cellular labelling with F-actin demonstrated that there was a reorganization of actin filaments and formation of stress fibres in the cells cultured in TGF-β2, these being also characteristics resulting from the EndMT process (represented by red fluorescence) (Figure 4).

Since molecular changes consistent with EndMT were observed, we decided to evaluate whether there are functional alterations in ECs after treatment with TGF-β2. Unlike mesenchymal cells, ECs are known to form a network of vessel-like structures when seeded onto matrigel in the presence of angiogenic growth factors. Upon TGF-β2 treatment, all ECs showed reduced capacity to form vessel-like structures, and this inhibitory effect was more pronounced in PAECs (Figure 5).

Upon ligand binding, TGF-β2 receptor complexes activate both Smad and non-Smad signalling pathways. In order to examine which of these signalling pathways are engaged in the TGF-β2-induced EndMT, we analysed the expression of phosphorylated Smad 3 by immunoblotting. However, the basal levels of phosphorylated Smad 3 did not increase upon TGF-β2 treatment in all endothelial cells studied (Appendix A).

Although TGF-β signaling occurs mainly via the Smad pathway, other pathways that are collectively referred to as ‘non-canonical’ can also be activated as a complement to Smad action. Since TGF-β2 treatment induced morphological and molecular changes consistent with EndMT without enhancing the phosphorylation of Smad 3, we investigated whether this process was mediated by activation of Smad-independent signalling pathway(s).

We assessed the phosphorylation status of eight proteins that could be activated through Smad-independent signalling in cell lysates treated with TGF-β2 for five days.

Among the evaluated proteins, we only observed increased levels in the phosphorylated ERK 1/2 upon TGF-β2 treatment and this occurred exclusively in PAECs. There was a 2.5-fold increase in the fluorescence mean intensity (FMI) of phosphorylated ERK1/2 in PAECs after EndMT induction (Figure 6A). These results were further confirmed by immunoblotting (Figure 6B). We tested the inhibition of ERK1/2 by adding 1 µM of U0126 inhibitor in the control and TGF-β2-treated PAECs (after 15 and 30 min of treatment). We observed that chemical inhibitors of MEK1/2 (U0126) prevented the TGF-β2-induced phosphorylation of ERK1/2 in the PAECs treated with TGF-β2 in both time periods (Figure 6C). These results suggest that the activation of EndMT in PAECs may have been mediated by the activation of the ERK pathway.

## 3. Discussion

EndMT is an important biological process that occurs during embryonic development and some cardiovascular pathologies. The evidence obtained in this study demonstrates that ECs isolated from different anatomical sites behave differently during TGF-β2-mediated activation of EndMT.

It is known that EndMT activation is crucial to cardiac development, and TGF-β2 plays an important role during this process. Azhar et al. demonstrated that the inhibition of TGF-β2 in mice prevents cardiac EndMT and, consequently, the development of the heart, whereas knockout mice for TGF-β1 or TGF-β3 have normal development [18,19].

Due to the importance of TGF-β2 during cardiac development, we initially studied its role in cardiac ECs from two different arteries (PAECs, derived from the aortic artery and CAECs, derived from the coronary artery).

After five days of treatment, there was an insignificant downregulation of the endothelial marker CD31 in both PAECs and CAECs. However, only PAECs underwent a clear morphological transition to a mesenchymal-like shape. In addition, it was found that these cells acquired a gene expression signature more consistent with EndMT (increase of mesenchymal markers *SM22*, *CNN1*, *ACTC1*, *Col1A1*, and transcription factor *SLUG*).

In agreement with our results, Kokudo et al. observed the differentiation of embryonic stem cells -derived ECs (MESECs) in murine cells after treatment with TGF-β2, with increase in the expression of the markers SM22α, calponin (CNN1) and Snail1 [7].

In addition, Arcieniegas and coworkers showed that adult bovine endothelial cells could differentiate into smooth muscle cell-like cells in vitro. After induction with TGF-β1, these cells started to express α-SMA, whereas the expression of factor VIII was decreased [20].

Thus, it was observed that although CAECs and PAECs originate from the same organ (heart), the activation of EndMT after treatment with TGF-β2 was more evident in PAECs. Subsequently, the response of TGF-β2 to other types of ECs was investigated. ECs derived from the umbilical cord vein (HUVECs) and derived from the pulmonary artery (HPAECs) were used. Regarding these ECs, no morphological change consistent with EndMT after treatment was observed. In addition, the decrease of the CD31 endothelial marker was observed only in the HUVEC, but there was only increase in the expression of the *ACTC1* gene in the induced HUVECs. Regarding HPAEC, no decrease in CD31 was observed after treatment with TGF-β2, and a significant increase in the expression of the *ACTC1*, *SM22α*, *CNN1*, and *SNAIL* genes was observed.

However, the level of phosphorylated Smad3 remained unchanged in all TGF-β2-treated ECs lines compared to the controls, suggesting that there may be a TGF-β2 pathway independent of SMAD, causing EndMT mainly in PAECs.

Thus, it was shown that each type of ECs has different degree of plasticity when induced to the EndMT process in vitro, since the ECs from different anatomical sites evaluated in this study showed different responses when treated with TGF-β2.

In the same organ, the endothelium of small and large vessels, veins and arteries, exhibits significant heterogeneity. Thus, differences of responsiveness among ECs are expected, since the endothelium has a high degree of molecular, biochemical, and functional heterogeneity according to the organ’s origin and the microenvironment [21,22,23]. Moreover, due to differences in embryonic origin and structural and hemodynamic characteristics, it is not surprising that ECs from different vessels exhibit local morphological and functional specializations. Chi et al. performed a global gene expression analysis on ECs from different blood vessels and microvascular ECs from different tissues. It has been demonstrated that all these ECs have a distinct and characteristic gene expression profile. Specific differences in the pattern of gene expression distinguished EC from large vessels of microvascular ECs [4].

These differences may also occur with respect to the expression of some receptors among the ECs, such as expression of TGFβ receptors. The TGF-β ligand binds to a complex composed of two types of receptors-type I (TβRI), also known as activin receptor-like kinases (ALK), and type II (TβRII). There are seven different type I receptors (ALK1 to ALK7), however TGF- signalling in EC happens mostly via ALK1 and ALK5, along with TβRII [24]. Once TGF-β binds to the TβRII, it phosphorylates the components of the TβRI leading the signal propagation. In our study, gene expression of TβRII was previously analysed by qPCR in PAEC, CAEC, HUVEC, and HAEC. The analysis of qPCR, showed that PAEC and CAEC presented a greater expression of the TβRII gene in relation to the other ECs (Appendix A). We also observed that there were no significant changes in TβRI (ALK5) and TβRII proteins in all ECs in response to 48 h TGF-β2 treatment (Appendix A and, again, the PAECs and CAECs showed greater protein expression mainly of TβRII in comparison to the other ECs. These findings may explain the fact that PAEC and CAEC were the type of EC that underwent a more consistent response to EndMT after treatment with TGF-β2. Doerr and colleagues examined the expression of bovine aortic endothelial cells markers and TGFβ receptors in the presence or absence of low (0.5 ng/mL) and high (5 ng/mL) TGFβ2 concentrations, and under normoxic and hypoxic culture conditions. The cells expressed ALK5 (TβRI), TβRII, and endoglin (the endothelial-specific TβRIII), and no significant changes in these proteins were observed in response to 6-h TGFβ2 treatment in either normoxia or hypoxia [25]. Although we only addressed to ALK5 in this study, others TBRI receptors could influence on the EndMT process, such as ALK1, which is predominantly expressed in endothelial cells. As far as we know, no study has attended to this matter.

In our work, among the ECs induced by TGF-β2, HPAECs maintained CD31 expression even after treatment. In one study with ECs derived from the adult human and ovine aortic valve, Paranya et al. showed that these cells acquired a mesenchymal phenotype when treated for six days with TGF-β1. Double labelling of the endothelial marker CD31 and the mesenchymal α-SMA marker were observed in these ECs treated both in vitro and in vivo. However, throughout the experiment, expression of the CD31 protein was maintained in cells that activated EndMT, suggesting that the process was incomplete and that the experimental conditions were not sufficient to allow the reduction of this endothelial marker. Notably, performing the same treatment on HUVECs and human dermal microvascular endothelial cells (HDMECs) did not trigger EndMT [26].

We observed that the level of phosphorylated Smad3 remained unchanged in the four TGF-β2-treated ECs as compared to the controls. Since apparently the canonical pathway of TGF-β did not mediate EndMT activation after treatment, it was evaluated which Smad-independent pathway could be involved in the phenotypic change of cells that underwent EndMT. Among the signalling pathways investigated, the activation of the MAPK-Erk1/2 pathway was observed in the treated PAECs. The increase in ERK1/2 phosphorylation after treatment with TGF-β2 in these cells was corroborated by Western blot.

While Smad-dependent TGF-β signalling has been extensively studied in EMT and EndMT processes, roles for ERK in TGF-β signalling, specifically in the context of these processes, have been slowly elucidated [27].

Regarding EndMT, its induction by TGF-β has been examined in many distinct cultured ECs, but few studies focus on possible Smad independent pathways that act in this process. In addition, it is noted that the majority of works involving the non-canonical pathway of TGF-β, mainly ERK, are related to EndMT that happens during development or cardiac pathogenesis.

In a study, it was shown that ERK phosphorylation is required for EndMT in the atrioventricular cushion in the development of murine embryos [28].

It is known that TGF-β can stimulate multiple intracellular signalling pathways, but it is not clear which of these pathways direct aortic diseases. A study has shown that the activation of ERK 1/2 by TGF-β contributes to the progression of aortic aneurysm in murine model of Marfan syndrome, and that antagonists of this pathway may be therapeutically advantageous [29].

Ghosh et al. showed that murine cardiac ECs underwent TGF-β2-mediated EndMT through canonical Smad signalling and the envelopment of the non-canonical ERK pathway. Results revealed that after the use of inhibitors, the phosphorylation of SMAD2 and ERK 1/2 was inhibited, as well as the activation of EndMT [30].

In another study, it was demonstrated that TGF-β1-induced EndMT in ECs of the heart valve and in carotid artery is accompanied by the increase in Erk phosphorylation. The direct blockade of phosphorylated ERK was sufficient to prevent EndMT in cardiac ECs. The ability of ERK inhibitors to block EndMT suggests that these drugs may be useful in the manipulation of this process by preventing fibrosis occurring soon after myocardial infarction [31].

Medici et al. demonstrated that TGF-β2 stimulates EndMT via Smad-dependent and independent signalling pathways (MEK/ERK signalling pathways, PI3K and p38 MAPK) in human dermal microvascular endothelial cells (HCMECs). Inhibitors of this pathway prevented TGF-β2-induced EndMT [16].

In addition, the inhibition of endogenously activated signals by TGF-β in ECs by a small molecule that inhibits TGF-β kinase receptors leads to a decrease in the EndMT process, suggesting that the inhibition of TGF-β signaling can suppress EndMT [32]. Zeisberg et al. demonstrated that systematic administration of recombinant human BMP-7 (rhBMP-7) significantly reduced EndMT during cardiac fibrosis [9]. However, further studies are needed to identify the molecular mechanisms by which BMP-7 inhibits EndMT.

Therefore, TGF-β stimulated EndMT has been examined in several types of cultured ECs, and in each cell type, this process is activated by different signalling pathways. This suggests that the type of EC and the microenvironment influence the signalling pathway used to initiate EndMT.

The identification of the mechanisms that control EndMT is important for translational applications in clinical medicine. Such knowledge may be beneficial for the design of therapeutic strategies against diseases associated with EndMT such as fibrosis, cancer, diabetes, and atherosclerosis.

## 4. Material and methods

### 4.1. Cell Cultures

We used distinct types of endothelial cells (ECs): CAEC (coronary artery endothelial cells, ATCC^®^ -Catalog No. PCS-100-020), PAEC (aortic artery endothelial cells, ATCC^®^ -Catalog No. PCS-100-011), HPAEC (pulmonary artery endothelial cells, ATCC^®^ -Catalog No. PCS-100-022) and HUVEC (human umbilical vein endothelial cells). HUVEC were obtained from human umbilical cords collected at the Reference Maternity Hospital (MATER, Ribeirão Preto, Brazil) after obtaining informed consent from all the mothers. This study was approved by the Reference Maternity Hospital (MATER process number 027/2011), and by the Ethical Committee of the School of Medicine of Ribeirão Preto (protocol n 1157/2012, 27 June 2012). All ECs were maintained in EGM (endothelial growth medium)-2 (Lonza, Walkersville, MD, USA), containing 5% fetal bovine serum (FBS) (Hyclone, Logan, UT, USA) and 1% penicillin/streptomycin.

### 4.2. HUVEC Isolation and Culture

Briefly, the umbilical cord was collected in sterile buffer solution containing NaCl 1.4M, Na_2_HPO_4_ 0.01 M, KH_2_PO_4_ 0.001M, KCl 0.04M, 0.02% glucose, and 10% of albumin (pH 6.5). After cannulation, 10 mL of buffer solution were used to wash out the blood inside the vein (approximately 10 cm in length). Then, the extremity was clamped and the umbilical cord was filled with collagenase type IA (0.5%) digestion (Sigma, St. Louis, MO, USA) for 20 min at 37 °C. The collagenase solution with the detached cells was harvested, and the vein was washed twice with RPMI (5% of FBS) (Gibco, Waltham, MA, USA). After centrifugation at 250× *g* for 10 min, EC pellet was resuspended in 10 mL of EGM2 (Lonza, Walkersville, MD, USA) with 5% of FBS. Cultures were maintained at 37 °C in humidified CO_2_ incubator (Thermo Scientific Pierce, Rockford, IL, USA). Media were changed each 48 h of initial plating until the confluence was reached. After expansion, the cells of the third passage were analyzed by flow cytometry (FACsort model, BD Biosciences, CA, USA).

### 4.3. Flow Cytometric Analysis

Cell samples (5 × 10^5^ cells) were incubated for 15 min with fluorochrome-conjugated antibodies (5 μL) against CD31, CD90, CD144, CD105, CD146, CD34, CD73, CD140b, CD29, CD13, CD166, CD44, CD49e, CD54, CD51/61, and CD14 (BD Biosciences, San Diego, CA, USA) or isotype matched controls. Subsequently, the cells were washed by centrifugation at 300× *g* and resuspended in PBS. Cells were collected (10,000 events) in FACS Calibur (BD Biosciences, CA, USA) and analysed using CellQuest software (BD Biosciences, CA, USA).

### 4.4. Matrigel Assay

Three-dimensional cultures were established by plating different endothelial cells on Matrigel (BD Biosciences, CA, USA). According to the manufacturer’s recommendations, 350 μL of Matrigel (3.43 mg) was used in a 24-well plate (1.75 cm^2^ per well). The entire procedure was performed on ice and after the application of Matrigel, the plate was incubated at 37 °C for 30 min to solidify the Matrigel. Approximately 7 × 10^4^ cells were resuspended in 650 μL of the EGM-2 culture medium with 5% FBS and added to the well. Cells were cultured in 5% CO_2_ at 37 °C for 24 h. After 24 h of incubation, the cultures were observed under the inverted optical microscope to monitor the formation of capillary-like structures.

### 4.5. TGF-β2 Treatment and Erk Kinase 1/2 Inhibition

ECs were grown in culture using EGM-2, containing 5% FBS and 1% penicillin/streptomycin, followed by human endothelial serum-free medium (ESFM) (Gibco, Grand Island, NY, USA) 24 h prior to all experimental conditions. Recombinant TGF-β2 (R&D Systems, Minneapolis, MN, USA) was added to the serum-free culture medium for all relevant experiments at a concentration of 10 ng/mL during five days. The medium exchange and TGF-β2 addition were performed daily for five days. Small-molecule inhibitors were added to cultures 15 and 30 min prior to treatment with TGF-β2. The MEK (MAPK/ERK kinase)1/2 inhibitor U0126 (Sigma, St. Louis, MO, USA) was used at a concentration of 1 μM.

### 4.6. RNA Isolation and RT-PCR

Total RNA was prepared using RNeasy Reagent (Qiagen, Hilden, Germany) and reverse-transcribed by random priming and using High Capacity cDNA Reverse Transcription Kit (Life Technologies, Woolston, UK). Quantitative RT-PCR was conducted using the TaqMan^TM^ Master Mix kit (Life Technologies, Woolston, UK) and analysis was performed using the 7500 Real TimePCR system (Life Technologies, Woolston, UK). Gene expression data was normalized by β-actin and β2-microglobulin. Data analysis was performed using the comparative Ct (ΔΔ*C*t) quantitation method [33]. For primer sequences, see Table 1.

### 4.7. Immunofluorescence Staining

Endothelial marker was assessed by immunofluorescence using an antibody against CD31 (BD Biosciences Pharmigen, San Diego, CA, USA). All primary antibodies were revealed with a secondary antibody conjugated anti-mouse IgGs with Alexa Fluor 488 (Invitrogen Molecular Probes, Eugene, OR, USA). Nuclei were stained with dihidrocloreto de 4,6 diamino-2-fenilindol; (DAPI) (Invitrogen Molecular Probes, Eugene, OR, USA).

### 4.8. Western Blotting

Cells were washed in cold PBS and lysed with RIPA buffer (Thermo Scientific) containing Halt phosphatase and proteinase inhibitor cocktail (ROCHE, Mannheim, Germany) according to the manufacturer’s protocol. Three sonication cycles were carried out of 5 min each in an ultrasonic bath (Unique, São Paulo, SP, Brazil) with cooled water. Lysates were centrifuged at 20,000× *g* for 30 min at 4 °C, and the supernatants were designated as total cell lysates. Protein concentration was determined using BRADFORD Protein Assay kit (Bio-Rad, Hercules, CA, USA). Equal amount of protein (30 μg) was loaded in each well of 10% Tris-glycine gel (Bio-Rad) and subjected to electrophoresis. Proteins were transferred to polyvinylidene fluoride membranes (GE Healthcare, Chicago, IL, USA) and then blocked with 5% non-fat dry milk in Tris-buffered saline (TBS) followed by overnight incubation with primary antibodies at 4 °C. Membranes were washed in TBS containing 0.05% Tween 20. Corresponding horseradish peroxidase (HPR)-conjugated anti-rabbit or anti-mouse IgGs (Cell Signalling, Danvers, MA, USA) were used as secondary antibodies. Blots were probed with an anti β-actin or GAPDH antibody as a loading control. Western blotting analysis was performed with the following antibodies using dilutions and protocols recommended by the respective manufacturers: anti-phospho-ERK1/2 (ab4819-Abcam, Cambridge, UK), anti-ERK1/2 (M 5670-Sigma), anti-phospho-SMAD3 (ab51451- bcam), anti-SMAD3 (#9513-Cell Signaling), anti-CD31 (#3528-Cell Signaling), anti-SM22-alpha (ab14106-Abcam), GAPDH (#2118-Cell Signaling), and anti-β-actin (sc-81178-Santa Cruz, Dallas, TX, USA). The protein-antibody complex was detected using the ECL Western Blotting Detection Reagent (GE Healthcare, Chicago, IL, USA,) and signals were detected using a CCD camera (Image Quant LAS 4000 mini) (GE Healthcare, Chicago, IL, USA).

### 4.9. Milliplex Map Assay

It was used MILLIPLEX Kit™ MAP 8-plex Multi-Signalling Pathway (Millipore, cat. no. 48-680MAG), which is based on Luminex xMAP^®^ technology (Luminex, Austin, TX, USA). This kit was used to detect phosphorylated proteins from: ERK/MAP kinase 1/2 (Thr185/Tyr187), STAT3 (Ser727), JNK (Thr183/Tyr185), p70 S6 kinase (Thr412), IkBa (Ser32), STAT5A/B (Tyr694/699), CREB (Ser133), and p38 (Thr180/Tyr182) pathways. The protein extraction and detection procedure were performed according to the manufacturer’s instructions.

### 4.10. Statistical Analyses

All data were expressed as the mean ± S.D. values determined from three independent experiments. Results were compared by Student’s *t*-test. Differences were considered significant when *p* < 0.05. All statistical tests were two-sided. These statistical analyses were performed using the GraphPad Inc. Prism 6 version 6.01 software.

## 5. Conclusions

The data obtained in this study showed that TGF-β2 induces EndMT in vitro in different types of ECs. Among the macrovascular ECs analysed in this study, PAECs were the most responsive to TGF-β2 treatment, presenting phenotypic changes consistent with the in vitro EndMT process. The changes observed in these cells were probably initiated by the activation of the Erk1/2 pathway.

Thus, although there are similarities between the EndMT and EMT (epithelial mesenchymal transition) process, since many molecular mechanisms act in both processes, there are different and specific mechanisms involved during EndMT activation that can be dependent on the anatomical origin of the ECs. Since this process may cause or aggravate some pathological conditions such as fibrosis and tumour metastasis, the selective inhibition of the Erk pathway could suppress these pathologies induced and aggravated by TGF-β2.

## Figures and Tables

**Figure 1 ijms-20-00458-f001:**
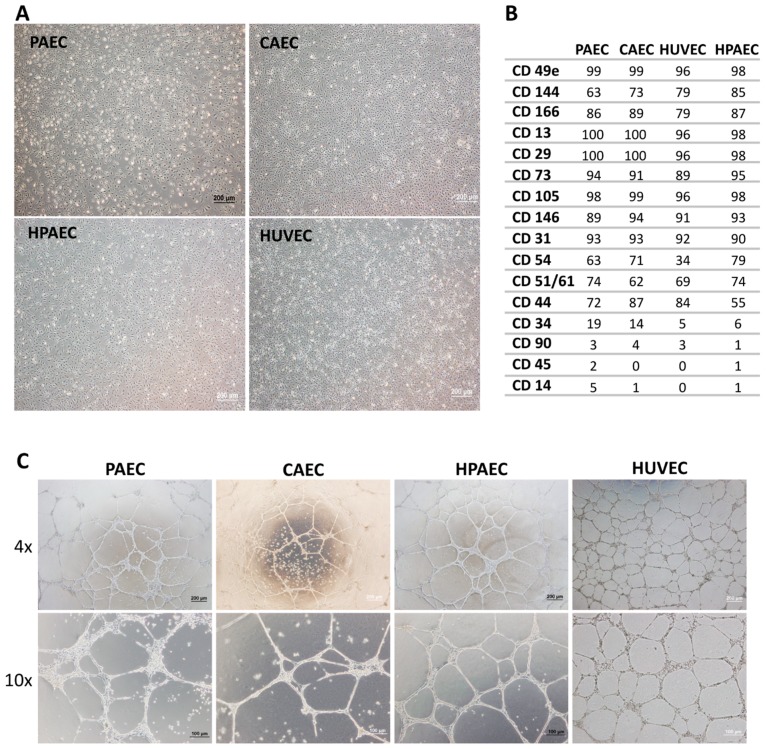
Endothelial cells characterization regarding morphological, immunophenotyping, and vessel-like structures assay. (**A**) Phase contrast micrography demonstrating the polygonal morphology of aortic artery endothelial cells (PAEC), coronary artery endothelial cells (CAEC), human umbilical vein endothelial cells (HUVEC), and pulmonary artery endothelial cells (HPAEC) cells (100× magnification). (**B**) Immunophenotyping of ECs by flow cytometry. (**C**) All endothelial cells (PAEC, CAEC, HUVEC, and HPAEC) were able to form vessel-like structures when grown in matrigel, evidencing characteristics typical of CEs (40× and 100× magnification).

**Figure 2 ijms-20-00458-f002:**
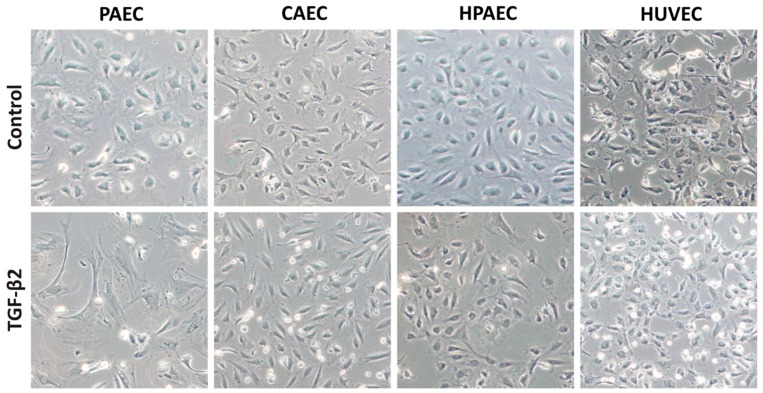
Phase contrast micrographs of endothelial cells after five days of TGF-β2 (transforming growth factor β2) induction. Morphological alteration was observed only in PAECs treated (100× magnification).

**Figure 3 ijms-20-00458-f003:**
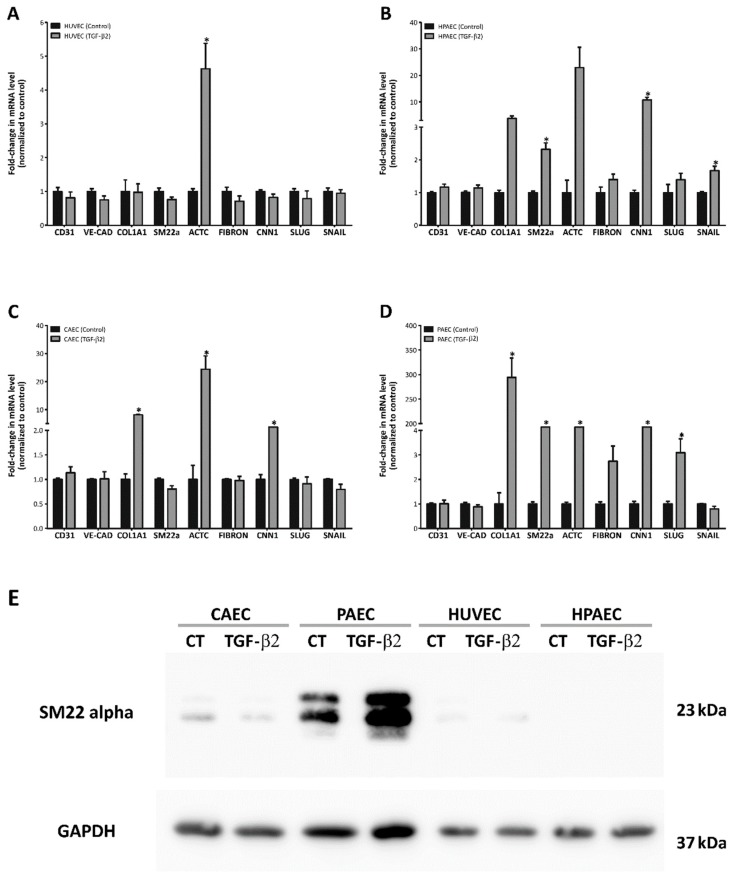
Molecular changes observed after EndMT induction in different endothelial cells. (**A**–**D**) Analysis of the expression of the endothelial markers (*CD31* and *VE-cadherin*), mesenchymal markers (*COL1A1*, *SM-22α*, *ACTC*, *Fibronectin* and *CNN1*), and transcription factors (*SLUG* and *SNAIL*) by real-time PCR (*n* = 3, * *p* < 0.05; of Student). (**E**) Protein analysis by Western blot of the mesenchymal marker SM-22α. GAPDH was used as endogenous control (representative image of one replicate).

**Figure 4 ijms-20-00458-f004:**
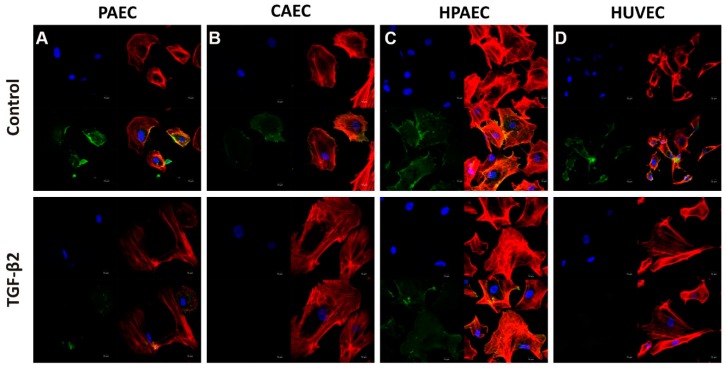
Characterization of EndMT induction by TGF-β2 (10 ng/mL) in cell lines (**A**) PAEC, (**B**) CAEC, (**C**) HPAEC, and (**D**) HUVECs (non-treated or treated with TGF-β2). Immunofluorescence microscopy of cell lines induced to EndMT shows a decrease in the fluorescent intensity of CD31 (green) in PAECs, CAECs, and HUVECs cells. The nuclei were stained with DAPI (blue) and F-actin were stained with Phalloidin (red) (scale bar 50 µM; representative image of one replicate of each sample).

**Figure 5 ijms-20-00458-f005:**
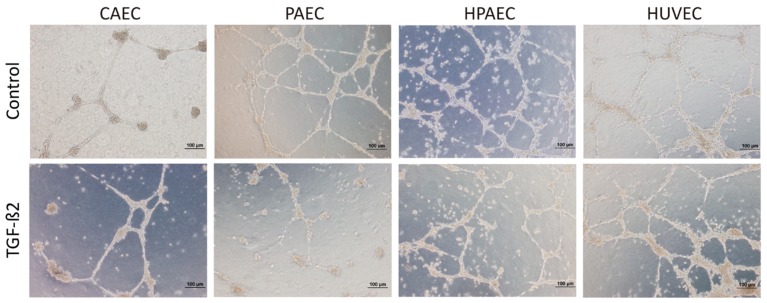
TGF-β2 decrease formation of vessel-like structures in the cell lines (CAEC, PAEC, HPAEC, and HUVEC). The cells were treated with TGF-β2 and evaluated the capacity formation of vessel-like structures. This inhibitory effect was observed mainly in PAECs (representative image of one replicate; *n* = 3).

**Figure 6 ijms-20-00458-f006:**
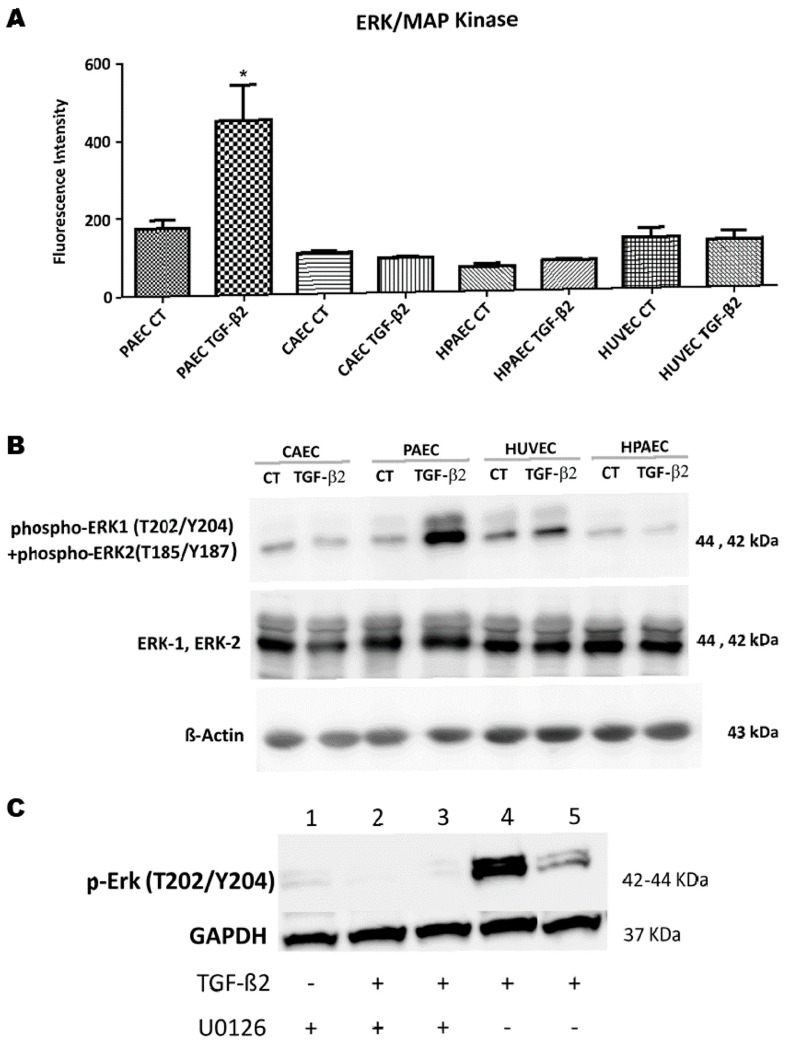
Effect of EndMT on the activation of the Erk pathway. The cells (CAEC, PAEC, HUVEC and HPAEC) were cultured for five days in presence TGF-β2 (10 ng/mL). Aliquots were withdrawn after the treatment and evaluated by (**A**) Multiplex technique analysis Array Kit (*n* = 3, * *p* ≤ 0.05) and (**B**) western blotting using phospho-Erk1/2 (Thr202/Tyr204) and ERK1/2. β-actin were used as endogenous controls (representative image of one replicate of each sample). (**C**) Chemical inhibitor against MEK1/2 (U0126; 1 μM) inhibits the increase of ERK1/2 phosphorylation in the PAECs treated with TGF-β2. 1) U0126; 2) U0126-15′ TGF-β2; 3) U0126-30′ TGF-β2; 4) TGF-β2-15′; 5) TGF-β2-30′. GAPDH were used as endogenous controls (representative image of one replicate of each sample).

**Table 1 ijms-20-00458-t001:** Probes used in qPCR reactions and their respective target genes.

Genes	Code
*PACTB (β-actin)*	4326315E
*ACTC1 (Actin, alpha cardiac muscle 1)*	Hs01109515_m1
*CDH5 (VE-cadherin)*	Hs00174344_m1
*CNN1 (Calponin 1)*	Hs00154543_m1
*COL1A1 (Collagen type 1)*	Hs00164004_m1
*FIBRONECTIN*	Hs01549976_m1
*PECAM-1or CD31 (platelet/endothelial cell adhesion molecule)*	Hs00169777_m1
*SLUG*	Hs00950344_m1
*SNAIL*	Hs00195591_m1
*TAGLN ou SM22-α (Smooth muscle protein 22-alpha)*	Hs00162558_m1
*β2-MICROGLOBULIN*	4333766

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
