# Peer review of "Endothelial Cells Tissue-Specific Origins Affects Their Responsiveness to TGF-β2 during Endothelial-to-Mesenchymal Transition"

_ijms, 2019, doi:10.3390/ijms20030458_

Reviewer 1 Report

In the paper Ferreira and colleagues investigate the contribution of anatomical origin of endothelial cells on EndMT in response to TGF-b2. For the purpose they used four differente macrovessel endothelial cell models, HUVEC, CAECs, PAECs and HPAECs and found that PAECs were the most responsive to TGF-b2 and acquired the mesenchymal phenotype in response to the cytokine.

Concerns:

1.      Why is the TGF-b2 used at 10 ng/ml? Why the authors did treat the cells for 5 days? Is the protocol justified by previous results?

2.      Fig. 2: Are the results reproducible? The picture are representative of how many experiments? Is the profiferation of cells affected ? Quantification in 5 random field of cell diameter would help to appreciate the difference.

3.      Line 213: the authors state that the transcription level of CD31 in cells exposed to TGF-b2 is not affected, however they did not show any results. Please show the QPCR of CD31.

4.      Fig. 4: this figure need to be modified: the four pictures that form each panel for each cell type and experimental condition need to be showed and described  in figure legend, separately (in the present form it is not clear that each panel is formed by 4 pictures). Are the results reproducible? Please indicate the number of replicates.

5.      Fig. 3, 5 and 6: The number of replicates need to be reported.

Minor:

1.      In table 1 change colagen 1 with collagen 1 and VE caderin with VE cadherin

2.      Figure legends of figure 5 and 6 need to be revised

3.      Line 278-279, 297, 298, 300, 305: change CEs with ECs

Author Response

Thank you very much for swift handling of our manuscript and for giving us the opportunity to resubmit.

We have addressed the reviewers’ comments by adding all required data to the manuscript as described in our point-by-point response letter. The changes introduced are marked by yellow color.

1. The starting concentration of 10 ng/mL for TGF--β2 for EndMT was based on previously published articles. We also tested other TGF-β2 concentrations in one of the ECs and found out that the cells responded better to 10 ng / mL than other concentrations, with a better established EMT (Epithelial Mesenchymal transition) model in mammary epithelial cells (MCF-10A). We performed tests with different incubation periods (48 hours and 5 days) and the results consistent to EMT were more evident with the longer exposition. Mesenchymal cells markers were increased in the treated cells, with decreased epithelial markers expression. These cells also had an exacerbated Snail 1 expression, a well-known EMT transcription factor. 

2. Figure 2 is representative of the 3 independent experiments, being therefore reproducible. We did not perform any assay that proved that cell proliferation is affected, but visually it appears that there is a slight delay in cell proliferation following treatment with TGF-β2.

3. The first group of bars in the bar graph of Figure 3 (A-D), show the qPCR result of CD31 gene expression. There was no significant change in this gene expression in either of the cells with or without TGF-β2 treatment. In order to highlight this important point, we included the referred figure citation, the citation in line 221 of the manuscript.

4. The Figure 4 as well as its legend have been modified (line 229):

Figure 4.Characterization of EndMT induction by TGF- β2 (10 ng/ mL) in cell lines (A) PAEC, (B) CAEC, (C) HPAEC and (D) HUVECs (non-treated or treated with TGF- β2). Immunofluorescence microscopy of cell lines induced to EndMT shows a decrease in the fluorescent intensity of CD31 (green) in PAECs, CAECs and HUVECs cells. The nuclei were stained with DAPI (blue) and F-actin were stained with Phalloidin (red) (representative image of one replicate of each sample).

5. As requested, the number of replicates were reported in the manuscript. Figure 3: line 217-219; Figure 5: line 242 and Figure 6: line 265-267.

 Minor:

1. Both words are changed in table (page 4).

2. The legend in figure 5 has already been revised and changed in the manuscript (line 240).

Figure 5. TGF-β2 decreased formation of vessel-like structures in the cell lines (CAEC, PAEC, HPAEC and HUVEC). The cells were treated with TGF-β2 and the capacity formation of vessel-like structures was evaluated. This inhibitory effect was observed mainly in PAECs (representative image of one replicate; n=3).

The legend in figure 6 has already been revised and changed in the manuscript (line 263)

Figure 6. Effect of EndMT on the activation of the Erk pathway. The cells (CAEC, PAEC, HUVEC and HPAEC) were cultured for 5 days in presence TGF-β2 (10 ng/mL). Aliquots were withdrawn after the treatment and evaluated by (A) Multiplex technique analysis Array Kit (n = 3, *P 0.05) and (B) western blotting using phospho-Erk1/2 (Thr202/Tyr204) and ERK1/2. β-actin were used as endogenous controls (representative image of one replicate of each sample).

3. These words have been changed in the mentioned lines.

Reviewer 2 Report

In this manuscript, the authors examined the effect of TGF-b2 on EndMT in several ECs and found that TGF-b2 treatment induced EndMT in PAECs more effectively than CAECs, HUVECs and HPAECs. Although the findings are interesting, I have several comments.

1. Responses to TGF-b2 are different among the examined ECs. Expression of TGF-b2 receptors should be examined in these cells.

2. Explanation for Figure 1 is too short. Please explain the findings shown in Figure 1 in more detail.

3. Figure 4. Please explain F-actin expression in the Results.

4. ERK1/2 and p-ERK1/2 expression was examined in cells treated with TGF-b2 for 5 days. If the authors want to say that the ERK1/2 signaling pathway is involved in TGF-b2-induced EndMT, ERK expression should be examined in cells soon after TGF-b2 treatment (several minutes and/or several hours).

Does treatment with an ERK pathway inhibitor suppress TGF-b2-induced EndMT?

Author Response

Thank you very much for swift handling of our manuscript and for giving us the opportunity to resubmit.

We have addressed the reviewers’ comments by adding all required data to the manuscript as described in our point-by-point response letter. The changes introduced are marked by yellow color.

1. Gene expression of TGF-β receptor 2 (TGFBR2) was previously analyzed by qPCR in all ECs used in this study (data not shown in the article). The analysis of qPCR, based on the calculation of relative expression units  (ALBESIANO et al., 2003), showed that PAEC and CAEC showed a greater expression of the TGFBR2 gene in relation to the other ECs.

ALBESIANO, E., et al. Activation-induced cytidine deaminase in chronic lymphocytic leukemia B cells: expression as multiple forms in a dynamic, variably sized fraction of the clone. Blood,102:3333-9, 2003.

2. “One of the major concerns in using primary cells in experiments is the degree of contaminants with other cell types. So, all cells types were properly characterized before performing each procedure. The EC population was obtained after four culture passages, when they presented a uniform and typically polygonal morphological appearance typical of CEs (Figure 1A). Based on data obtained after flow cytometric immunophenotyping, ECs expressed little or none of the CD34, CD45 and CD14 hematopoietic cell antigens, nor the typical CD90 mesenchymal antigen, whereas they were positive for endothelial markers such as CD31, CD144, CD105, CD146, among others (Figure 1B). In addition, the functional test of the EC cultures in matrigel, showed the ability of ECs to form complex structures similar to vascular networks, evidencing characteristics typical of CEs (Figure 1C).” This explanation will be added in the results section (line 177 - 186).

3. The paragraph containing the explanation about F-actin expression will be added in the results section (line 223 - 227): “Remodeling of actin filaments is necessary for EndMT. Cellular labeling with F-actin demonstrated that there was a reorganization of actin filaments and formation of stress fibers in the cells cultured in TGF-β2, these also being characteristics resulting from the EndMT process.”

4. We tested the inhibition of Erk1 / 2 by adding U0126 inhibitor (1 uM)  in the PAECs controls and PAECs treated with TGF-β2 (after 15 and 30 minutes of treatment). We observed that there was suppression of the Erk1 / 2 pathway in the PAECs treated with TGF-β2 in both time periods tested (data not shown in the article). For a clearer understanding see the the Western blot (Figure A).

Representative figure A (in below file):

U0126; 2) U0126 (15' TGF-b2); 3) U0126 (30' TGF-b2); 4) TGF-b2 (15'); 5) TGF-2b (30');

Round  2

Reviewer 2 Report

All comments have been addressed in Responses to Reviewer's Comments. However, revision is partial.

1. Differential responses of ECs to TGF-b2 should be discussed by citing related references, such as Albesiano et al., Blood 2003, about TGF-b receptors.

2. Experiments using U0126 are very important. Please include the data in the manuscript.

Author Response

1. Although TGF-β receptor 2 data were not shown in this manuscript, we will add in the discussion session an paragraph about the expression of this receptor in the ECs (highlighted in red color). We didn’t cite the suggested reference (Albesiano et al., Blood 2003), because this article is not about the TGF-β receptors, but rather about the relative expression units calculation used for the analysis of the qPCR of this receptor in the EC.

2. The data of experiment using U0126 inhibitor were included in the manuscript as Figure 6C. The results and material and methods related to this experiment were added in their due sessions highlighted in red color.